# Material Treatment in the Pulsation Reactor—From Flame Spray Pyrolysis to Industrial Scale

**Stefan Heidinger** [1,*] **, Felix Spranger** [1] **, Jakub Dostál** [1] **, Chunliang Zhang** [1] **and Christian Klaus** [2]

1. Boysen-TU Dresden-Research Training Group, Technische Universität Dresden, 01062 Dresden, Germany;
   felix.spranger@tu-dresden.de (F.S.); jakub.dostal@tu-dresden.de (J.D.); chunliang.zhang@tu-dresden.de (C.Z.)
2. IBU-Tec Advanced Materials AG, Hainweg 9-11, 99425 Weimar, Germany; klaus@ibu-tec.de
* Correspondence: stefan.heidinger@tu-dresden.de; Tel.: +49-351-463-43137

**Abstract:** Current challenges in the areas of health care, environmental protection, and, especially, the mobility transition have introduced a wide range of applications for specialized high-performance materials. Hence, this paper presents a novel approach for designing materials with flame spray pyrolysis on a lab scale and transferring the synthesis to the pulsation reactor for mass production while preserving the advantageous material properties of small particle sizes and highly specific surface areas. A proof of concept is delivered for zirconia and silica via empirical studies. Furthermore, an interdisciplinary approach is introduced to model the processes in a pulsation reactor in general and for single material particles specifically. Finally, facilities for laboratory investigations and pulsation reactor testing in an industrial environment are presented.

**Keywords:** mobility transition; pulsation reactor; flame spray pyrolysis; flame spray synthesis; material treatment; catalysts; exhaust gas treatment; interdisciplinary





## 1. Introduction

Applications for specialized high-performance materials range from emission-reducing catalysts in combustion engines to next generation filter materials. As the demand for such materials is expected to rise in the near future, their production should be scaled up accordingly. Although flame spray pyrolysis (FSP) is an efficient way to synthesize these materials primarily on the laboratory scale, pulsation reactor (PR) technology represents a promising mass-production method for meeting the needs of the mobility transition. Similar to FSP, treating precursors with the pulsating conditions inside PR results in special material properties, such as small particle sizes and highly specific surface areas. For emission-reducing catalysts, for example, these properties are especially advantageous. Thus, this work presents the approach of utilizing FSP for the investigation and synthesis of novel materials on a laboratory scale and transferring the findings with the aim of scaling the production using PR technology. A proof of concept was established by transferring single synthesis processes from the FSP to the PR through trial and error approaches, as presented below. Since such approaches are costly and time demanding, numerical models for the prediction of pulsation reactor behavior and of heat and mass transfer at the particle level need to be developed and derived from the theoretical and experimental findings. For any further development of this concept, a detailed understanding of the processes in both apparatuses is crucial, which is what this research aims for through a combination of experimental and theoretical investigations. Here, an interdisciplinary approach is applied involving chemistry, energy process engineering, and measurement technology, along with laboratory investigations and PR testing in an industrial environment.

First, the flame spray pyrolysis is introduced, followed by a presentation of the pulsation reactor technology, including the setup of a pilot scale PR. Then, proof of concept trials are presented and the shortcomings of this approach are discussed. Afterwards, we transition to an alternative approach in which mathematical models are applied in order

to link the constructional and operational parameters of a PR to the process conditions within and to link these process conditions to the processes at the level of a single particle under treatment. These models allow for the implementation of an initial step of theoretical investigation in order to conduct any subsequent experiments more deliberately. The presentation of the models is then complemented by the presentation of a lab scale PR with the attached measurement equipment. The measurement setup, which is tailored to the PR, is explained and the inferred knowledge from the measurements as they relate to the validation of the mathematical models and a deepening of the understanding of the processes in a PR is discussed.

## 2. Material Synthesis with Flame Spray Pyrolysis

Catalytically active materials have long been the focus of scientific and industrial research for a long time due to their applications in a plethora of fields. Many of those applications are of critical relevance to modern societal changes like the mobility transition, including durable and sustainable emission reduction catalysts, materials for efficient energy storage, or materials relevant for water splitting and the subsequent conversion of hydrogen in fuel cells. An interesting class of materials that can be applied for these purposes and which have garnered increasing interest in recent years are (mixed) transition metal oxides. These compounds, which include well-known materials, such as, for example, perovskites [1], are interesting due to their lower cost, higher availability, and lower susceptibility to poisoning compared to common catalysts that contain precious metals. They also exhibit high thermal-stability, among other advantageous features [2].

Many such compounds have already been investigated as catalysts for different applications. Regarding emission abatement, active precious metal-free catalysts have been tested for the oxidation of different pollutants, such as $NO_x$ [3], different hydrocarbons and volatile organic compounds (VOCs) [4–6], and carbon monoxide [7]. For example, Li et al. reported on the synthesis of Sr-doped lanthanum cobalt oxide perovskites and the higher $NO_x$ conversion rates achieved by them than common Pt-containing catalysts [8]. In order to synthesize those compounds on a laboratory scale, which is a vital first step for the discovery and development of novel materials, different approaches were chosen. Usually, facile synthesis paths such as the precipitation or sol-gel method are preferred.

Here, the advantage is that these are relatively easy and reproducible ways of generating nanoparticles with the desired compositions. However, they often lead to unfavorable impurities or material properties, for example a relatively small specific surface area. A promising way to synthesize metal oxides with desired properties is flame spray pyrolysis, the principle of which is shown in Figure 1. For this method, a salt or metal-organic precursor containing the chosen metal is dissolved in a combustible solvent, which is subsequently dispersed into fine droplets and ignited. A mixture of oxygen and methane is used as the flammable mixture, with additional oxygen being applied to disperse the liquid precursor. From the droplets, metal oxide nanoparticles are formed via various pathways. They can then be collected using a glass fiber filter to which negative gauge pressure is applied. The resulting powder can be used without the need for further treatment. The method is described in detail elsewhere [9,10]. Since this designation can be misleading, especially in an interdisciplinary context, for this publication the method will be referred to as flame spray synthesis (FSS). Via this method, the metal oxide nanoparticles reside in the flame just very shortly, up to 100 ms [11], but at temperatures higher than 2000 °C [12]. This ensures a formation of crystalline particles, but at the same time prevents strong agglomeration and sintering, as is the case during a calcination step, which is required in most sol-gel or precipitation approaches. This leads to a narrow size distribution and highly specific surface area for nanoparticles synthesized via FSS, which are important prerequisites for the use as catalysts. For example, FSS-made copper manganese oxides, so called hopcalites, reached far better CO conversion rates than commercial ones made by a precipitation method [13,14].

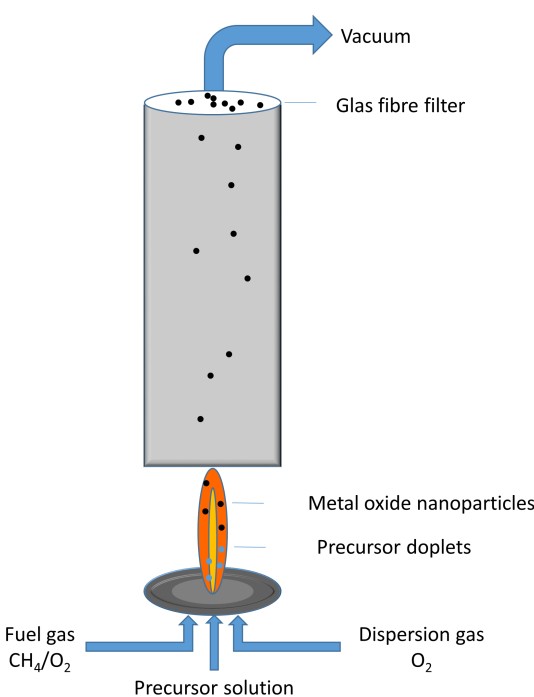

**Figure 1.** Principle of the flame spray synthesis.

However, FSS is not a suitable method when trying to meet the industrial scale demands of a high material throughput while maintaining the advantageous material properties. It is rather a tool to manufacture novel materials with the desired properties at a laboratory scale. At this stage, a different type of reactor, enabling material synthesis and treatment at a larger scale with results comparable to FSS, is introduced.

## 3. Material Synthesis in a Pulsation Reactor

A pulsation reactor (PR) is an apparatus designed for continuous thermal material treatment and production of fine powder materials utilizing a pulsating hot gas stream. Even though other sources are potentially possible, the pulsating flow in a typical PR, which is displayed in Figure 2, is generated by periodic combustion, often being referred to as pulsating combustion (see, for example, [15]).

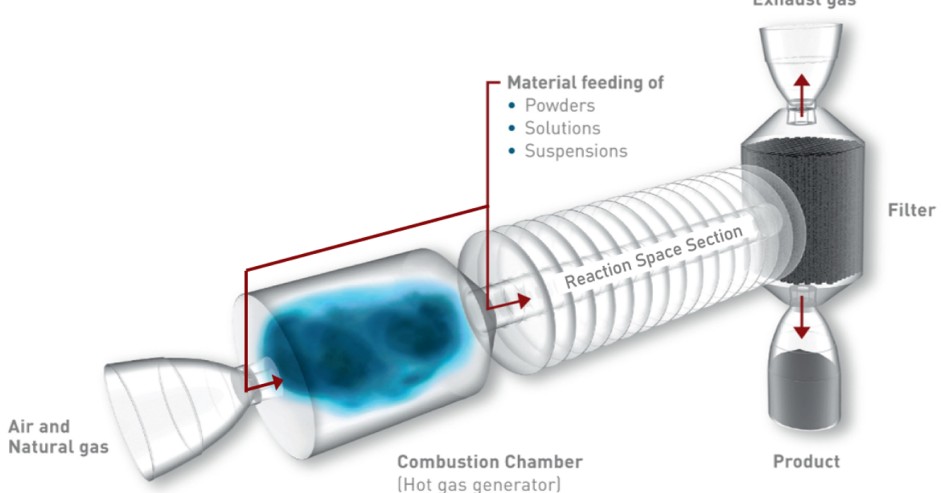

**Figure 2.** Pulsation reactor setup.

The working principle of a PR based on pulsating combustion is displayed in Figure 3. For a self-excited periodic combustion, a combustible gas mixture is driven through inlet

valves into the combustion chamber, where it is ignited. That generates an excess pressure which pushes the combustion products further into the tailpipe. The inertia of the outgoing gas causes a negative relative pressure to form briefly in the combustion chamber, which causes fresh fuel and air to be pulled into the chamber, while a portion of the flue gas is sucked back into the combustion chamber as well. The mixture is then reignited and the cycle starts over again. This sub-process is the basis for the underlying frequency of each pulsation reactor, which, depending on the system design and operating parameters, ranges from 1 to 500 Hz [16].

The precursors, which are introduced into the PR in the form of either a powder, solution, or suspension, are distributed in the flow to form an aerosol. The material is then treated while being transported by the pulsating hot gas stream. After the flow has emerged from the reaction space (typically the tailpipe, therefore, also called the reaction pipe), its temperature is reduced by introducing a cooling gas. The final product is then separated from the flow using an exhaust gas filter or a cyclone. The typical treatment temperature falls between 250 °C and 1300 °C (see [16] and Table 1), and the reactants in a pulsation reactor experience a thermal shock-like treatment with short residence times and characteristically fast heating and cooling rates. Pulsation reactors are typically used for the production of catalysts for industrial applications, battery materials, materials for electronic components, polishing agents, pigments, or materials for UV protection applications. In addition to material synthesis, pulsation reactors can also be applied for other thermo-chemical conversion processes, such as drying, calcination, and annealing.

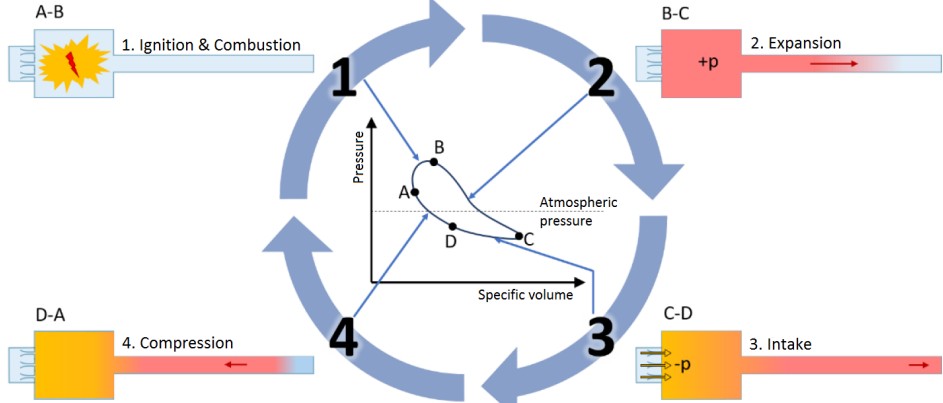

**Figure 3.** The pulsating combustion cycle [16]. *1. Ignition & Combustion*: combustible mixture is driven into the combustion chamber, where it is ignited; *2. Expansion*: the rapid increase in pressure accompanying the combustion blocks further combustible mixture from entering the PR and the combustion products are pushed out of the combustion chamber; *3. Intake*: inertia of the outgoing flue gas causes a negative gauge pressure to form in the combustion chamber, which sucks fresh combustible mixture inside; *4. Compression*: the negative gauge pressure causes a portion of the hot flue gas to travel back into the combustion chamber as well, which compresses the fresh combustible mixture and supplies thermal energy for re-ignition. The combustion cycle then repeats.

A pulsation reactor might essentially be perceived as a special form of an entrained flow reactor, while only differing in the form of the flow. A steady flow is present in an entrained flow reactor, in contrast to the pulsating flow in a PR. The incessant periodic variation in the relative velocity between the gas and the particles (i.e., the periodically varying Reynolds number) causes an increase in the mean Nusselt number and consequently results in an improved convective heat transfer [17]. In the work by Dec and Keller [18], the mean Nusselt number for a pulsating flow was reported to be up to 2.5 times larger compared to a steady flow of the same mean Reynolds number. However, Xu et al. [19] observed a heat transfer coefficient enhancement of 2 to even 5 times.

Although commercially used PRs do exist, two devices, dedicated for research, were constructed. One of them is a PR test rig for measurements (as addressed later), the other is a pilot plant PR, depicted in Figure 4. Both devices were built in accordance with theoretical and practical design studies [20]. The pilot reactor poses an intermediate step in the transfer of product developments from the FSS to industrial production. Several different geometries of the combustion chamber and resonance tube can be applied, leading to broad ranges of operational parameter values, as shown in Table 1. The coordinated modularity and adaptability of the pilot system not only drastically reduces the financial risks of scaling, but also allows a comparison of different system configurations with relatively little effort. Due to the design and size of the reactor, it delivers results from small quantities of raw material—an advantage leading to reduced test times and costs, especially with high-quality precursors. Even demanding material systems can be handled safely due to the system environment designed for this purpose.

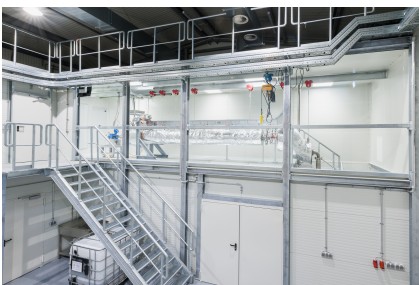 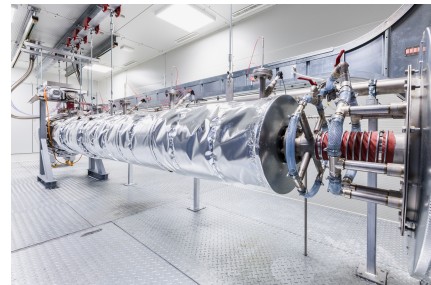

**Figure 4.** Pilot plant pulsation reactor (Kleinmengen-PR) at *IBU-tec Advanced Materials AG*, *Weimar*. View including the production hall (left) and a close-up view (right).

**Table 1.** Operational and constructional parameter ranges of the pilot scale pulsation reactor [16].

| Parameter | Range (Pilot Plant) | Often Applied |
|---|---|---|
| Fuel | natural gas | - |
| Thermal energy consumption | up to 70 kW | - |
| Temperature | 250 °C to 1300 °C | 450 °C to 950 °C |
| Pressure | up to 20 mbar | 3.5 mbar to 15 mbar |
| Residence time | 0.05 s to 2 s | 0.5 s |
| Reaction pipe length | 1.2 m to 7 m | 5 m |
| Flow velocity | 5 m/s to 20 m/s | 10 m/s |
| Throughput (raw material) | 0.1 kg/h to 20 kg/h | 3 kg/h |
| Product separation | cyclone and cartridge filter | - |

With this pilot plant PR and the presented FSS setup, empirical studies were conducted in order to check the possibility of transferring synthesis from the FSS to the PR.

## 4. FSS and PR Comparison—Empirical Studies

In the empirical studies, the goal was to produce particles similar to those derived from FSS in the PR. The materials chosen for these preliminary tests were zirconia ($ZrO_2$) and silica ($SiO_2$). Liquid Zr(IV)-propoxide was thermally treated in the PR directly while, for the FSS, the precursor was dissolved in 2-propanol, since Zr(IV)-propoxide is not combustible and, hence, not suitable for this kind of reactor in its pure form. The applied operating parameters are shown in Table 2. For both presented FSS syntheses, the fuel gas flows were 3 L/min oxygen and 1.5 L/min methane. The dispersion gas flow was 7.5 L/min, with a dispersion pressure drop of 2 bar.

**Table 2.** Comparison of operation parameters and process conditions at FSS and PR for the synthesis of $ZrO_2$.

|  | **FSS Reactor** | **Pulsation Reactor** |
|---|---|---|
| $ZrO_2$ content feed material | 1.5 mol/L | 2.2 mol/L |
| Solvent | 2-Propanol | none |
| Educt feed | 0.3 L/h | 3.9 L/h |
| Product throughput | 55 g/h | 1050 g/h |
| Residence time | approx. 0.1 s | 0.4 s |
| Absolute pressure | 1 bar | 1 bar |
| Temperature | >2000 °C | 1000 °C |
| Pulsation frequency | - | 20 Hz |
| Energy consumption | 47 MJ/kg$_{material}$ | 80 MJ/kg$_{material}$ |

To compare their properties, the resulting particles were examined by various analytical methods, including X-ray diffractometry (XRD), nitrogen-physisorption, and transmission electron microscopy (TEM) imaging. The diffractograms depicted in Figure 5 show that the crystalline phases are the same for both samples, matching the $ZrO_2$ reference plotted in red.

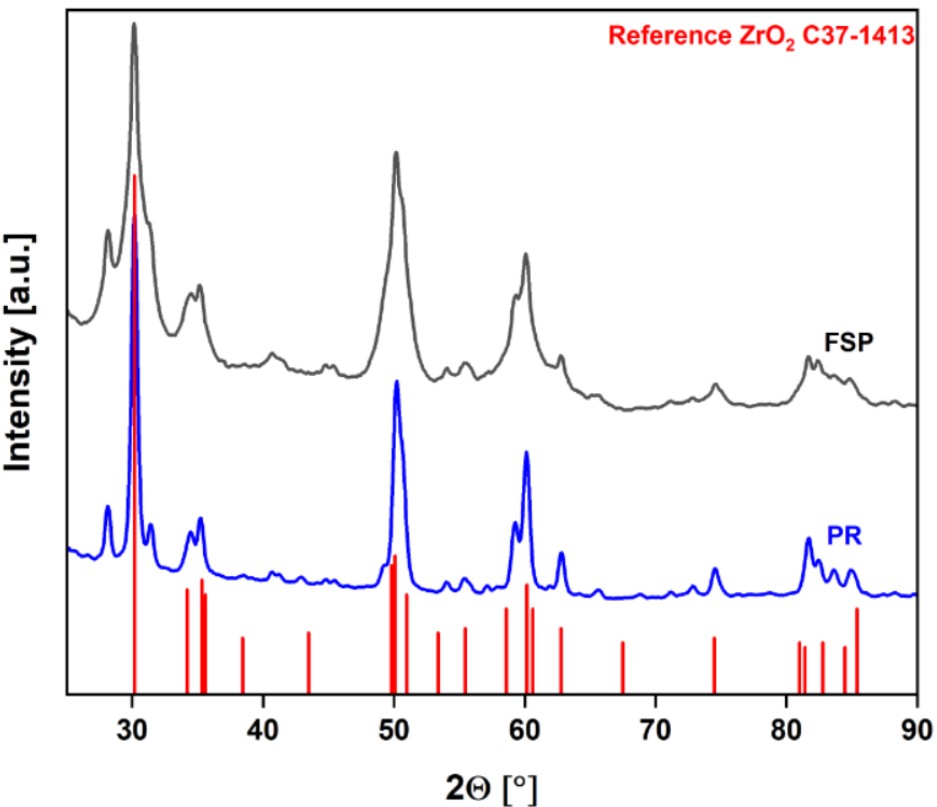

**Figure 5.** Diffractograms of the $ZrO_2$ samples from the FSS and PR.

The specific surface area (SSA) was measured via the BET method with a Quantachrome NOVA 4000 at −196 °C after activating the samples at 150 °C for 12 h in a dynamic vacuum. It was determined to be 66 m$^2$/g and 77 m$^2$/g for FSS and PR-made powders, respectively. Additionally, in the TEM images, the morphology of the powders appears similar: mostly small, spherical primary particles that are agglomerated. The pictures presented in Figure 6, however, indicate that the size dispersion of the PR particles is not as large as for FSS.

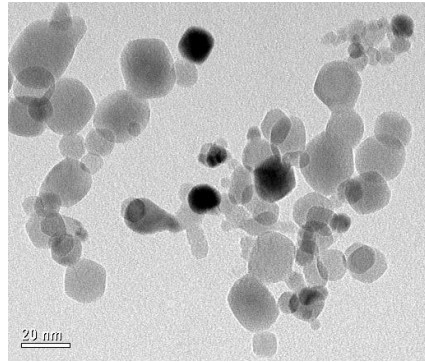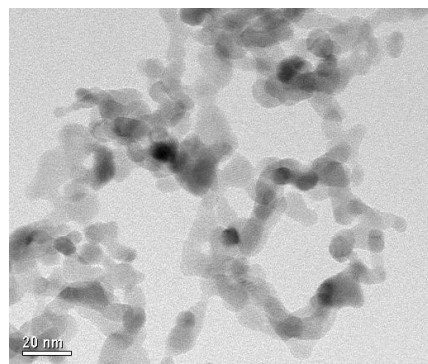

**Figure 6.** TEM images of ZrO$_2$ from FSS (**left**) and PR (**right**).

This is mainly due to the relatively high throughput of the FSS leading to larger particles with a smaller SSA. According to the material properties, very similar and comparable ZrO$_2$ nanoparticles were synthesized in the FSS and PR [21]. In addition, the desired ability to upscale material production can be highlighted, since an almost twenty-fold increase in material throughput without any decline of the product quality was achieved. Additionally, the use of an additional solvent was not necessary, which makes production more cost effective and sustainable. After conducting this successful trial, the production of SiO$_2$ was studied following the same procedure. Tetraethyl orthosilicate (TEOS) was used as a precursor. Again, 2-propanol had to be used as a solvent for the FSS, while in the PR it was applied purely. The hourly product throughput was comparable to the first trial. This and the used precursor concentrations are summarized in Table 3.

**Table 3.** Comparison of operating parameters and process conditions between FSS and PR for the synthesis of SiO$_2$.

|  | **FSS Reactor** | **Pulsation Reactor** |
|---|---|---|
| SiO$_2$ content feed material | 1.3 mol/L | 4.4 mol/L |
| Solvent | 2-Propanol | none |
| Educt feed | 0.3 L/h | 3.3 L/h |
| Product throughput | 55 g/h | 870 g/h |
| Residence time | approx. 0.1 s | 0.4 s |
| Absolute pressure | 1 bar | 1 bar |
| Temperature | >2000 °C | 1000 °C |
| Pulsation frequency | - | 20 Hz |
| Energy consumption | 47 MJ/kg$_{material}$ | 90 MJ/kg$_{material}$ |

As for ZrO$_2$, many characteristics of the resulting samples were similar. Both are amorphous, and the morphology shown in the TEM images depicted in Figure 7 resembles those of the previous samples, as expected. In this TEM measurement, even the particle size and overall optics are very similar between the different synthesis methods.

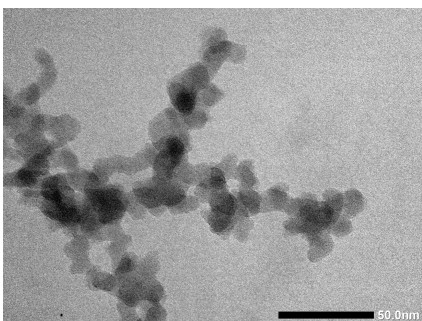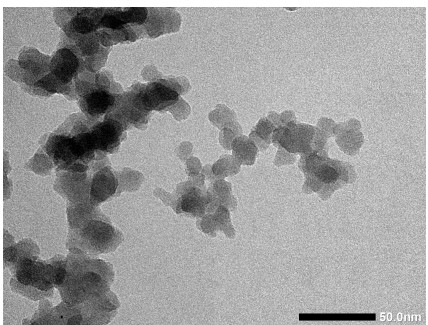

**Figure 7.** TEM images of SiO$_2$ from FSS (**left**) and PR (**right**).

In the case of SiO$_2$, however, there is a large difference concerning the SSA of the powders. The FSS sample exhibits an SSA of 260 m²/g, while the PR sample has less than half of that, at 114 m²/g. In both cases, ZrO$_2$ and SiO$_2$, the educt feed rate and the material throughput were similar. The same was true for the FSS. However, the material properties in the case of SiO$_2$, especially the SSA, differ strongly.

It is clear that these initial results have to be further examined and it has to be determine for which precursors and products the transfer is applicable. In addition, if the transfer from the FSS synthesis can be successfully understood in advance, this might enable feasibility studies for certain precursor systems. When compared to the efforts required for even a small-scale PR, substantial resources and time can be saved by holding these trials in a lab-scale FSS apparatus. Since the trials on the pulsation reactor were feasibility studies, little focus was placed on increasing throughput or reducing energy consumption. In larger campaigns, significant improvements can be expected in this respect. The two examples demonstrate the possibility of transferring synthesis processes from the FSS to the PR in principle. However, a deeper understanding, especially of the PR, is necessary to predict and tailor the product properties and to be able to correlate operational parameters of both apparatuses. Therefore, an interdisciplinary approach is now presented in order to model theses relations in a PR.

## 5. Interdisciplinary Approach

Now that the two apparatuses and their particularities have been introduced and the proof of concept has been demonstrated, an interdisciplinary approach for transferring the synthesizing process from the FSS to the PR is presented. This will involve chemistry, energy process engineering, measurement technology, as well as laboratory investigations and pulsation reactor testing in an industrial environment. This interdisciplinary approach is along the lines of suggestions from Meng et al. [22], who pointed out that studies on transferring pulsating combustion systems from laboratory scale to pilot and large scale should be carried out, while the exchange of gathered knowledge between academic institutions and industrial enterprises is recommended. When a material is synthesized in an FSS, the process conditions need to be captured to derive correlations between operating parameters and product properties. Meanwhile, the main focus with the PR lies on modeling and predicting the parameters of the internal processes. In order to examine the interrelationships of the material thermal treatment in the PR in more detail and to be able to compare it with the FSS, the system is divided into two sub-systems, as displayed in Figure 8. The influence of the PR design and operation parameters on the internal process conditions is considered separately from the influence of the process conditions on the single particles in a PR.

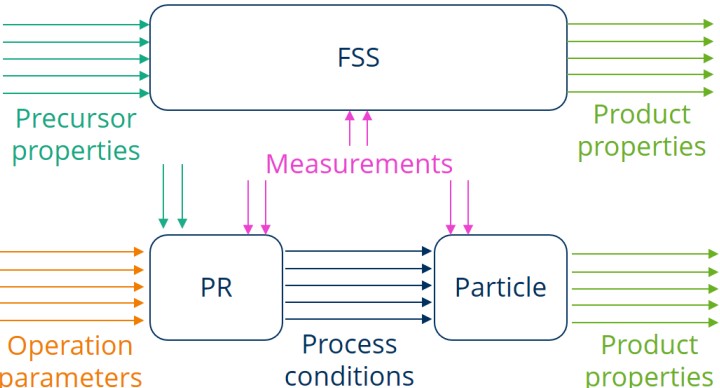

**Figure 8.** Approach applied in order to transfer the synthesis of novel materials from the FSS to the PR.

Additionally, Figure 8 also graphically represents the approach of transferring the material synthesis from the FSS to the PR. The steps that need to be carried out for that purpose are as follows:

1. New materials with advantageous properties are synthesized in the FSS, while the precursor properties and the process conditions leading to the product properties are determined;
2. The relations between the PR operation parameters and the resulting process conditions must be understood. For given operation parameters, the process conditions in a PR can then be predicted, for which purpose a numerical model might be used;
3. Similarly, the influence the process conditions have on the final product properties must be discovered. With these effects understood, the product properties might also be predicted;
4. Experimental data are required for an empirical investigation, as well as for validation of the theoretical models. In order to retrieve the parameters and conditions of interest, measurement setups must be designed for both the laboratory and the industrial environment and tailored to the specific conditions of the FSS and the PR;
5. The experimental and the theoretical findings and the developed models are to be transferred to pilot and industrial plants, which will help to upscale and optimize the material treatment process.

## 6. Process Model of a Pulsation Reactor

For an improvement in the design and operation of pulsation reactors, the relations between the design/operation parameters and the resulting process conditions need to be determined. Therefore, an analysis providing data from which broad conclusions can be drawn needs to be carried out. However, PRs have so far been designed and operated primarily by empirical knowledge and trial and error approaches [23,24]. Since this is both cost intensive and time consuming, a computational approach was taken here, with the development of a numerical model being the first stage.

Some models for pulsating combustion already exist. These might need further development and some may be used as a basis for a numerical model matching the needs of a proper PR analysis. The existing models are reviewed and their shortcomings are discussed next, which is followed by an outlook for the development of a pulsation reactor process model. A broad range of models for pulsating combustion have been developed, each of them considering the related processes and phenomena to a different level of detail. According to their spatial complexity, such models can generally be classified into two groups: zero- and one-dimensional models, and multidimensional models. On the one hand, zero- and one-dimensional (0D/1D) models are flexible and relatively easy to implement. They are able to simulate the basic features of pulsation reactors, e.g., the pressure and velocity oscillations or heat losses. Compared to multidimensional CFD models, they also require significantly less computational power and pre-processing efforts, e.g., geometry modeling or meshing. On the other hand, the spatial distribution of physical variables cannot be captured properly (1D models), or even at all (0D models), which does not allow for the investigation of complex flow patterns, temperature fields, flow-chemistry interaction, or flame structures. For such purposes, more advanced multidimensional CFD models have been developed.

The important 0D/1D pulsation reactor models are the AKT model by Ahrens et al. [25], the RMS model by Richards et al. [26], the BDB model by Barr et al. [27], and several alterations of these. The abbreviations always stand for the initials of the first three authors of the original model, and the key features of these models are summarized in Table 4.

**Table 4.** Overview of the 0D and 1D models for pulsating combustion.

| Model | AKT (1978) [25] | RMS (1993) [26] | BDB (1988) [27] |
|---|---|---|---|
| Spatial complexity | 0D | 0D | 1D |
| Balance equations | 2× 0D: energy balance for the combustion chamber; momentum balance for the tailpipe | 4× 0D: energy, mass, and species balance for the combustion chamber; momentum balance for the tailpipe | 3× 1D: momentum, continuity, and energy balance for the entire domain |
| PR type | Helmholtz type PR | Rijke type PR | Helmholtz type PR |
| Inflow modeling | Aerovalves modeled as flapper valves (having no backflow) | Continuous air and gas supply (no valves) | Time-varying mass flow rate originally defined as input |
| Combustion modeling | Combustion rate explicitly defined as constant | Single-step Arrhenius model | Released heat originally defined as input (results from experiments) |
| Thermal losses | Neglected | Explicitly defined heat transfer coefficient (combustion chamber only) | Explicitly defined overall heat transfer coefficient (convection at the inner PR wall, convection and radiation at the outer PR wall) |
| Further improvements | Inflow model without discontinuities; Explicitly defined overall heat transfer coefficient (convective and radiative heat loss from the combustion chamber); Friction modeled by a damping coefficient | No further improvements | Sub-models for inlet valves, combustion, heat transfer, and mixing |

The other group of models for pulsating combustion are the CFD models. Generally, they are all based on a system of standard balance equations. These are the conservation of momentum, mass, and energy for the mixture and, in case combustion is included in the model, conservation of mass for the individual constituents (i.e., conservation of species). A structured overview of the CFD models is provided in Table 5. All of the listed models focused on the Helmholtz type of PR and the simulations were always two-dimensional and axisymmetric.

**Table 5.** Overview of the CFD models for pulsating combustion.

| Author(s) | Computational Domain | Balance Equations | Turbulence Model | Combustion Model | Source of Pulsations / Inflow Modeling |
|---|---|---|---|---|---|
| Benelli et al. (1992) [28] | Inlet part, combustion chamber, tailpipe | Momentum, mass, and energy balance for the mixture; mass balance for individual components | k-ε, ASM | Single-step Arrhenius model | Flow-combustion interaction / Constraint between pressure and velocity, specified pressure loss coefficient |
| Möller and Lindholm (1999) [29] | Inlet part, combustion chamber, tailpipe, decoupler | Momentum, mass, and energy balance for the mixture; mass balance for individual components | LES | Two-step Westbrook–Dryer model | Explicitly specified in boundary condition / Specified mass flow rate |

**Table 5.** *Cont.*

| Author(s) | Computational Domain | Balance Equations | Turbulence Model | Combustion Model | Source of Pulsations | |
|---|---|---|---|---|---|---|
| | | | | | Inflow Modeling | |
| Tajiri and Menon (2001) [30] | Inlet part, combustion chamber, tailpipe | Momentum, mass, and energy balance for the mixture; mass balance for individual components | LES | Single-step Arrhenius model | Flow-combustion interaction | |
| | | | | | Specified stagnation pressure and temperature, inflow adjusts naturally | |
| Liewkongsataporn (2006) [31] | Tailpipe, decoupler | Momentum, mass, and energy balance for the mixture | V2F | Combustion not included | Explicitly specified in boundary condition | |
| | | | | | Specified total pressure | |
| Thyageswaran (2004) [32] | Tailpipe, decoupler | Momentum, mass, and energy balance for the mixture | k-ε | Combustion not included | Explicitly specified in boundary condition | |
| | | | | | Specified mass flow rate | |
| Zhonghua (2007) [33] | Inlet part, combustion chamber, tailpipe | Momentum, mass, and energy balance for the mixture; mass balance for individual components | k-ε | Single-step Arrhenius model | Flow-combustion interaction | |
| | | | | | Dynamic mesh for the flapper valve | |
| Yufen et al. (2013) [34] | Inlet part, combustion chamber, tailpipe | Momentum, mass, and energy balance for the mixture; mass balance for individual components | k-ε | Single-step Arrhenius model | Explicitly specified in boundary condition | |
| | | | | | Specified mass flow rate | |

Although several CFD models of a pulsation reactor have been developed so far, only a portion of those are capable of predicting the working conditions without requiring some experimental data. In this sense, coupling a model for the inlet valves to the conditions inside the pulsation reactor appears to be the biggest challenge. Therefore, the inlet condition is prescribed explicitly in a majority of the models, which directly defines the pulsation frequency. Only the models by Benelli et al. [28], Tajiri and Menon [30], and Zhonghua et al. [33,35] treat the inlet condition in a way which allows the formation of oscillations with a frequency corresponding to the combustion kinetics.

The advantages and drawbacks of the reviewed mathematical models (see Tables 4 and 5) were considered and two parallel approaches were selected. One approach is based on a 1D model and a self-developed computational tool, while the other one uses a 3D model and commercial CFD software. Although the output of the 1D model is a single value of each variable at an arbitrary coordinate along the PR, the 3D model is capable of delivering detailed fields of physical variables within the computational domain.

A detailed description of the models is not within the scope of this publication. However, their potential capabilities and drawbacks can be deduced from the previous section dedicated to the review of pulsating combustion models. Key aspects include the coupling of the flow through the inlet valves (focusing particularly on aerodynamic valves) with the conditions inside the PR, modeling of reaction kinetics, and heat transfer characteristics.

Although the 1D and the 3D models can work independently, they do not have to compete with each other, as the benefits of both can be effectively combined. The 1D model could be used to determine certain operating conditions, e.g., the frequency or the inflow over time, which afterwards could be explicitly applied to the 3D model in order to reduce the computational time. Irrespective of its type, the results of the PR process model can afterwards also serve as input parameters for a particle model, which is addressed in the following section.

### 7. Heat Transfer at Particles in Steady and Pulsating Flows

In addition to a model to link the operational and constructional parameters to the expected process conditions in a PR, considerations of the behavior of single particles under these conditions are necessary. In particular, the enhanced heat transfer, and in extension mass transfer, needs to be examined closely in the PR and compared with the FSS. With this in mind, this section provides a qualitative comparison of convective heat transfer of solid particles in steady and pulsating flows.

Entrained flow reactors (EFR) feature a steady flow $u_g = \bar{u}_g$, while in a pulsation reactor this steady flow is superimposed by a harmonically oscillating flow [36] $u_g = \bar{u}_g + U_g \cos(\omega t)$, with the pulsation frequency $\omega$ and the pulsation amplitude of $U_g$. For the sake of comparison, the flame spray synthesis is considered as an EFR here. It is theoretically demonstrated that a PR can be operated in such a manner that it matches the initial heat transfer of particles in an EFR. The quantity of motion, which translates to the convective heat transfer, is the slip velocity $u(t) = u_g - u_p$ between the gas and the particle. Therefore, this is the focus of the following considerations. When a particle is released in a steady flow $u_g = \bar{u}_g$ with a different velocity than the particle velocity itself, the particle will adapt to the velocity of the flow due to drag. This behavior is displayed in Figure 9a, where the slip velocity approaches zero while the particle velocity approaches the steady gas velocity.

In a pulsating flow the particle also tries to adapt to the velocity of the flow, which, in this case, is continuously changing. This is indicated by the angular frequency of the pulsation $\omega$. After the initial adaption to the flow, the particle velocity oscillates harmonically and provides a continuous slip velocity, which does not decay over time. This behavior can be seen in Figure 9b. Another beneficial phenomenon of pulsating flows is the enhanced initial slip velocity when releasing the particles into the reactor. By matching the time of release with the phase angle in the pulsation with the highest gas velocity, the initial slip velocity can be significantly higher than the mean flow velocity. This phenomenon is displayed in Figure 10.

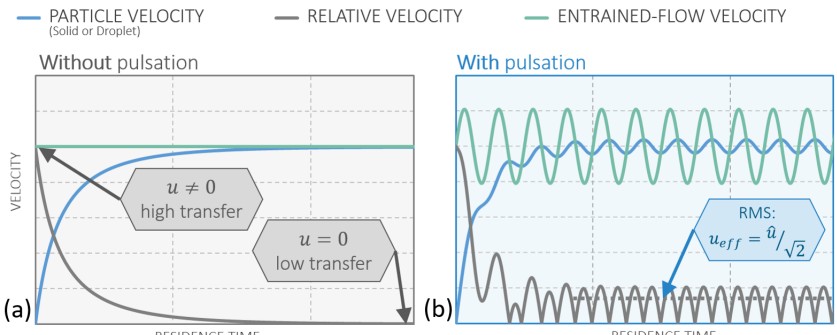

**Figure 9.** General plot of particle velocity, relative (slip) velocity, and gas velocity along the reactor (**a**) without pulsation (EFR); (**b**) with pulsation (PR).

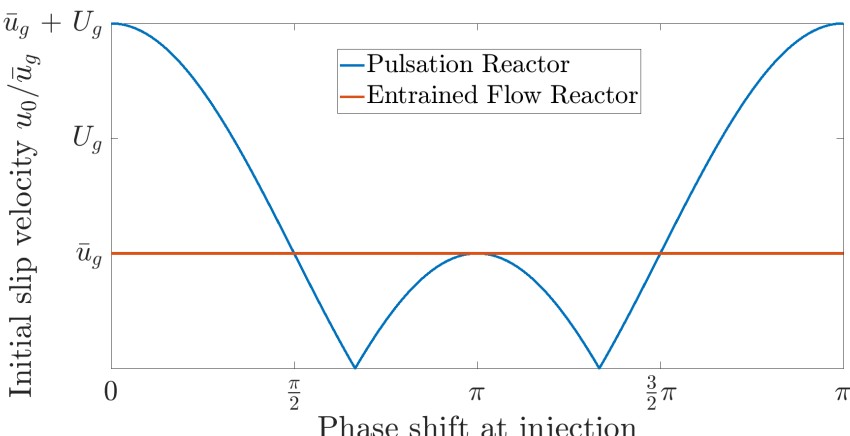

**Figure 10.** Initial slip velocity between gas and particle in respect of the phase shift at injection.

Heat transfer and, due to the Reynolds analogy, mass transfer can be modeled by applying appropriate empiric relations. Applying such a specific model for heat transfer is omitted here in order to focus on the concept itself, but all of the mainstream models predict an increase in heat transfer via the Nusselt number $Nu(u(t))$ with an increased slip velocity. Since the Nusselt number increases with increasing slip velocity, the pulsation leads to an improved heat transfer [17]. The criterion for the speed of the actual heating of the particle is the relaxation time $\tau_T = d^2 \rho_p c / 12\lambda$, with $d$ being the diameter, $c$ being the specific heat capacity of the particle, and $\lambda$ being the thermal conductivity of the gas. The particle temperature can be calculated with [37]:

$$\frac{dT_p}{dt} = \underbrace{\frac{T_g - T_p}{\tau_T}}_{\text{larger in FSS}} \underbrace{\frac{Nu(t)}{2}}_{\text{larger in PR}} \tag{1}$$

The gas temperature $T_g$ in the FSS is substantially higher (up to 2700 °C at the flame [12]) than in a PR (up to 1300 °C, as shown in Table 1). This relates to the first term in Equation (1) being in general larger in the FSS. In contrast, due to the increased slip velocity, the second term in Equation (1) is larger in the PR. Assuming the particles have the same properties ($d, \rho_p, c, T_{p,0}$) in both apparatuses, the process conditions in the PR ($\bar{u}_g, U_g, \omega, T_g$) can be set within their constructional limited range, as seen in Table 1, in order to match the temperature curve of the particles initially, as seen in Figure 11.

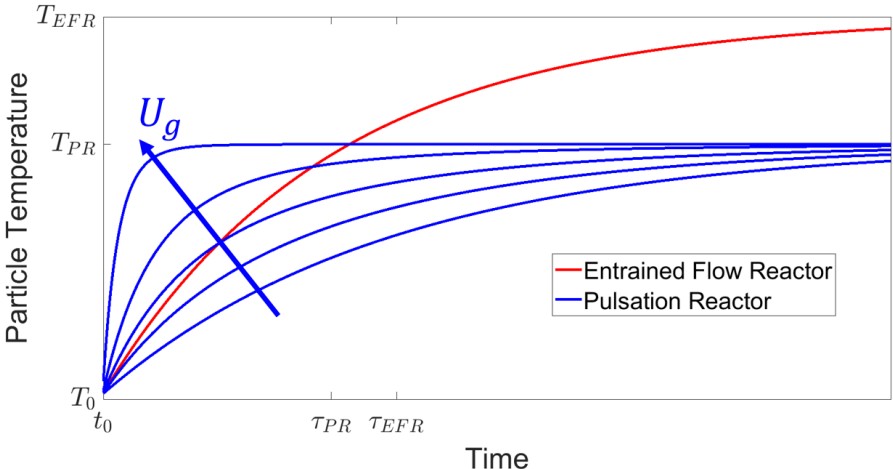

**Figure 11.** Initial particle temperature curves for PR and EFR.

The heat transfer in the PR from the hot gas to particles due to radiation is small (up to 3% of overall heat transfer [38]) and, besides the gas composition, primarily depends on the temperature $T_g$. With an increasing gas temperature, the radiative heat transfer increases as well. Therefore, in this case, its adjustability is very limited, but since it is expected to be larger in the FSS due to the higher gas temperatures, it can be accounted for by overcompensating the convective heat transfer in the PR.

## 8. Measurement and Model Validation

In order to gain a deeper understanding of the processes in the PR and to provide a method for validating theoretical models, experimental investigations are required. In addition to temperature and pressure, gas composition, flow velocity, flow pattern, and slip velocity between particles and gas need to be determined. For these experimental investigations, a reactor test rig was built at the Chair of Energy Process Engineering, TU Dresden, which is displayed in Figure 12. This test rig is equipped with extensive sensor technology. In Table 6, relevant measured quantities are assigned to the corresponding measurement methods and the inferred knowledge.

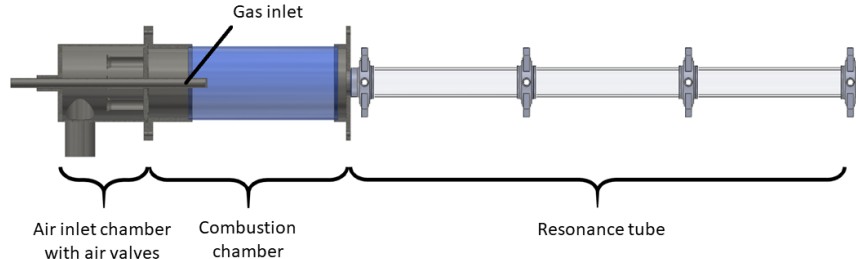

**Figure 12.** Design of the pulsation reactor test rig at the Chair of Energy Process Engineering, TU Dresden.

The central component of the multi-sensor measurement system is the particle image velocimetry system (PIV) for flow visualization and measurement. The basic design of such a PIV system is shown in Figure 13. This is a non-invasive method widely applied in current flow measurements and can be used to obtain instantaneous velocity profiles in high velocity flows and combustion measurements. In addition, PIV is also an appropriate method for validating CFD models since it can approximately measure the same level of detail that CFD predicts.

A prerequisite for PIV measurements is that the fluid under investigation contains tracers which scatter the light of a laser sheet. Images of this scattered laser light are taken in a short time sequence with a high-speed camera, from which the movement of the tracers within a known time difference is determined by means of cross-correlation. From these images, the flow direction and velocity are determined. The tracer particles must be large enough to scatter the laser light and small enough to follow the flow well. Green laser light with a wavelength of 532 nm is particularly bright and is, therefore, preferred for PIV measurements [39]. Since the particle size is limited downwards due to the Rayleigh scatter regime [39], tracer particles with a diameter from 0.5 μm to 2 μm are used for this application. Another limiting condition is the high-temperature of up to 1300 °C in the PR, which must not affect the integrity of the particles. Therefore, ceramic materials (e.g., $TiO_2$, $SiO_2$, $ZrO_2$, $Al_2O_3$) with significantly higher melting points are preferable [40]. A refractive index as high as possible and a low density that leads to a short relaxation time (low influence of inertia) are also criteria for particle selection [41,42].

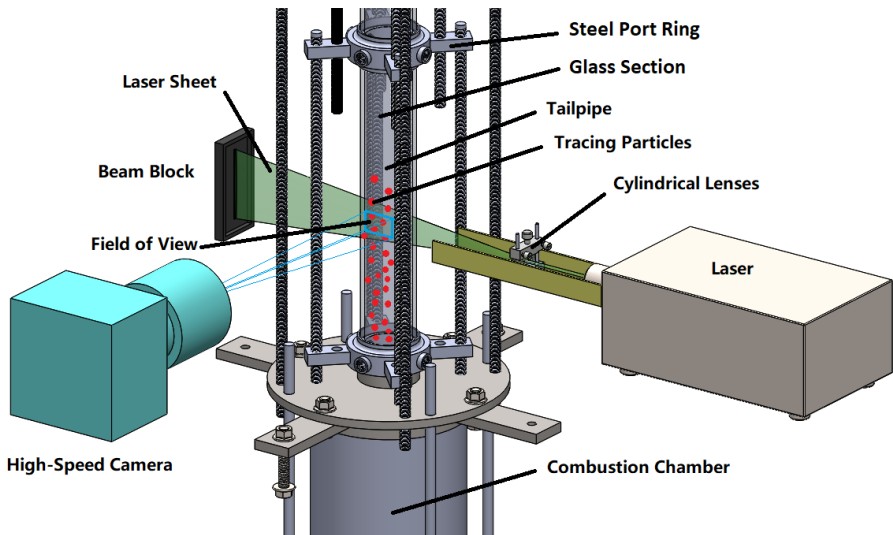

**Figure 13.** Working principle of the PIV system on the pulsation reactor.

In addition to any considerations of the particles, a distinction has to be made between continuous and pulsed lasers. The provision of pulsed laser light is significantly more cost intensive, but for the pulsed laser the time difference is determined by the temporal distance between the pulses and not by the frame rate of the camera, which makes it possible to detect higher velocities-up to 20 m/s for the PR. Preliminary considerations based on calculations according to Adrian and Yao [43] have shown that a continuous wave laser with a power of more than 5 W would be possible, but a pulsed laser is used in order to obtain more precise data and, if necessary, to increase reserves with regard to the maximum flow velocity. The components of the measuring system have to withstand or be protected from the high prevailing temperatures. The camera and laser optics are, therefore, located sufficiently far from the resonance tube where they are also protected by a cooling air flow. With this setup, flow measurements are possible at different angles and heights. The knowledge of the flow velocity distribution and the flow pattern in the resonance tube (an extension provided by the PIV measurements) are crucial in order to derive proper models for the process conditions in the PR. This ties into fundamental considerations, such as the necessity for multi-dimensionality of the model or the need to incorporate turbulence via sub-models.

In addition to the PIV system, other sensors are necessary, most of which require insertion openings. For this reason, the resonance tube is divided into glass sections for optical access and steel port rings which house temperature and pressure sensors. Thermocouples are particularly suitable for measuring high temperatures [44] and, consequently, used here to capture the gas temperature. The temperature profiles allow conclusions about the heat loss, which is essential for an energy balancing.

The periodically varying pressure is sampled with heat resistant microphones at much higher rates than the pulsation frequency, taking into account the sampling theorem. The amplitude of the pressure oscillations and their decrease over the resonance tube can be extracted from these data. The Fourier transform provides information on the oscillation frequencies from these time-resolved pressure measurements. Based on this, it is possible to analyze to what degree the pressure oscillations can be considered harmonic and can be modeled that way. In addition, as model validation, it will also be used to explain to what extent the predicted pressure distribution is in agreement with the measurement.

**Table 6.** Measurement quantities and methods as well as inferred knowledge for experimental investigations of the pulsation reactor.

| Quantity | Measurement Method | Measurement Position | Inferred Knowledge |
|---|---|---|---|
| Flow velocity | PIV | Glass sections of tailpipe | (1) Amplitude of velocity oscillations<br>(2) Degree of harmonic behavior of velocity<br>(3) Model validation |
| Flow pattern | PIV | Glass sections of tailpipe | (1) Degree of turbulence<br>(2) Degree of rotational symmetry in the flow<br>(3) Change in boundary layer compared to steady flow |
| Temperature | Thermo couple | Steel port rings | (1) Density of gas<br>(2) Heat losses<br>(3) Input for energy balance<br>(4) Model validation |
| Pressure | Microphone | Steel port rings | (1) Amplitude of pressure oscillations<br>(2) Frequency of pressure oscillations<br>(3) Degree of harmonic behavior of pressure<br>(4) Model validation |
| Exhaust gas composition | FTIR | Steel port rings | (1) Input for mass balance<br>(2) Occurring chemical reactions |
| Particle | Extraction | Steel port rings | (1) Secondary testing<br>(2) Occurring chemical reactions |
| Natural gas inflow | Flow sensor | Before combustion chamber | (1) Input for energy balance<br>(2) Input for mass balance |
| Air inflow | Flow sensor | Before combustion chamber | (1) Input for energy balance<br>(2) Input for mass balance |

The gas composition measurement, which is achieved by Fourier-transform Infrared Spectroscopy (FTIR), leads to a closed mass balance together with the inlet mass flow rates of air and natural gas, as well as the knowledge of the input flow and composition of the precursor solution. This balance enables a conclusion of any possible chemical reactions that may have occurred. Since the gas composition can be measured at several points along the resonance tube, a progression of these reactions can also be inferred. This is closely connected to the extraction of the product particles at several points along the resonance tube in order to analyze them via the same techniques as mentioned above and make inferences regarding existing reactions at the particle level.

## 9. Conclusions

A novel development process for materials with advantageous properties, such as small particle sizes and highly specific surface areas, was presented. As the first step, flame spray synthesis (FSS) was used to design and produce novel materials at the laboratory scale. Since the production process can not be upscaled via the FSS effectively, the pulsation reactor (PR) was introduced as an alternative together with an approach for transferring the material production from the FSS to the PR without affecting the product properties significantly. In this regard, the setups of a pilot scale and a laboratory scale PR were presented. Proof of concept was provided by two trials, in which zirconia and silica were synthesised via the FSS and the pilot scale PR, delivering products with similar particle sizes and specific surface areas. However, since these trial and error approaches are costly and time demanding, further investigations and a deeper understanding are required in order to control and optimize the production process. Pre-existing mathematical PR models were reviewed and their advantages and shortcomings were discussed. Based on this analysis, a hybrid approach featuring a one-dimensional and a three-dimensional model was introduced. It was theoretically demonstrated through heat transfer considerations

at the particle level that the improved heat transfer in a pulsation reactor can compensate for the low gas temperatures in the flame spray synthesis when treated as an entrained flow reactor. Therefore, initial heat transfer rates in the FSS can be matched by creating suitable process conditions in the PR. The measurement setup on a laboratory scale PR was explained and the inferred knowledge gained by the measurements in order to achieve a deeper understanding of the involved processes and to validate theoretical models was discussed.

**Author Contributions:** Conceptualization, investigation and writing—original draft preparation: S.H., F.S., J.D., C.Z. and C.K.; Writing—review & editing: S.H. and J.D. All authors have read and agreed to the published version of the manuscript.

**Funding:** The authors would like to thank the Boysen-TU Dresden-Research Training Group and IBU-tec Advanced Materials AG for the financial support that has made this publication possible. The Research Training Group is co-financed by TU Dresden and the Friedrich and Elisabeth Boysen Foundation. Grant number: BOY-135.

**Institutional Review Board Statement:** Not applicable.

**Informed Consent Statement:** Not applicable.

**Data Availability Statement:** Data are available from the corresponding author upon reasonable request.

**Acknowledgments:** This research is conducted as a joint project at Technische Universität Dresden with the Chair of Energy Process Engineering (M. Beckmann), the Chair of Inorganic Chemistry I (S. Kaskel), and the Chair of Magnetofluiddynamics, Measurement and Automation Technology (S. Odenbach).

**Conflicts of Interest:** The authors declare no conflict of interest.

## Abbreviations

The following abbreviations are used in this manuscript:

| | |
|---|---|
| AKT | Zero-dimensional PR model by Ahrens, Kim, and Tam [25] |
| BDB | One-dimensional PR model by Barr, Dwyer, and Bramlette [27] |
| BET | Brunauer–Emmett–Teller |
| CFD | Computational Fluid Dynamics |
| EFR | Entrained Flow Reactor |
| FFT | Fast Fourier Transformation |
| FTIR | Fourier Transformed Infrared Spectroscopy |
| FSP | Flame Spray Pyrolysis |
| FSS | Flame Spray Synthesis |
| PIV | Particle Image Velocimetry |
| PR | Pulsation Reactor |
| RMS | Zero-dimensional PR model by Richards, Morris, Shaw et al. [26] |
| SSA | Specific Surface Area |
| TEM | Transmission Electron Microscopy |
| UV | Ultraviolet |
| VOCs | Volatile Organic Compounds |
| XRD | X-ray Diffractometry |

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
