# Peer review of "Material Treatment in the Pulsation Reactor—From Flame Spray Pyrolysis to Industrial Scale"

_sustainability, doi:10.3390/su14063232_

Round 1
Reviewer 1 Report
This manuscript mainly focuses on Pulsation Reactor Technology. Some improvements are needed, for example:
- It’s better to improve the writing of this manuscript to meet the standard of scientific papers.
- This manuscript should focus more on the pulsation reactor technology, the contents related to other technology can be put into supplement information.
- The figures can be better organized. The permission should be got if the figures are from other papers.
- Some of the TEM images are not very clear.
- The format of some references should be revised.
Author Response
Stefan Heidinger
Technische Universität Dresden
TU Dresden-Boysen-Research Training Group
Chair of Process Engineering
01062 Dresden
25/02/2021
Dear Madam or Sir,
Thank you for your comments on our manuscript “Advancements in Material Production using Pulsation Reactor Technology”. In the following, we will address each of your comments:
- It’s better to improve the writing of this manuscript to meet the standard of scientific papers.
- The writing of the manuscript was improved, while a professional native speaker proofread the manuscript suggesting many changes, most of which were incorporated.
- This manuscript should focus more on the pulsation reactor technology, the contents related to other technology can be put into supplement information.
- The title of the manuscript was changed to “Material Treatment in the Pulsation Reactor - From Flame Spray Pyrolysis to Industrial Scale” in order to better display the focus of the manuscript not only on pulsation reactor technology, but rather the transfer of the synthesis process from the flame spray synthesis in the lab to the pulsation reactor for mass production.
- The figures can be better organized. The permission should be got if the figures are from other papers.
- The organization of the figures was improved, ensuring that the figures were always referenced before actually appearing. Some figures were corruptly displayed due to formatting issues. All figures were created by the authors.
- Some of the TEM images are not very clear.
- Due to a relatively low acceleration voltage of the used microscope (120 kV), it is difficult to get very sharp images of such small particles. The main messages that are important, like the particle size and morphology, are sufficiently recognizable, however.
- The format of some references should be revised.
- The format of all sources was revised, with some sources being updated in order to provide all necessary details.
Thank you for helping to improve this manuscript.
Sincerely,
Stefan Heidinger
Reviewer 2 Report
The article „ Advancements in material production using pulsation reactor technology, presents a very interesting approach to the new process of developing materials with favorable environmental properties, although it seems to be very costly and labor-intensive.
A question arises regarding the disposal of the resulting materials. The presented tests are performed under laboratory conditions but show great potential for the future. I find the job very interesting. It is presented in a clear, aesthetic, and legible way.
In my opinion, the work presented in this way is complete and does not require supplementing the content at the moment.
Author Response
Stefan Heidinger
Technische Universität Dresden
TU Dresden-Boysen-Research Training Group
Chair of Process Engineering
01062 Dresden
25/02/2021
Dear Madam or Sir,
thank you reviewing our manuscript.
Sincerely,
Stefan Heidinger
Reviewer 3 Report
Paper presents technologies for production of high-performance materials used in specialized application such as emission reducing catalysts in combustion engines.
The principles of material synthesis with Flame Spray Pyrolysis (FSP) / Flame Spray Synthesis (FSS) and Pulsation Reactor (PR) were presented as well as shortcomings and advantages were discussed.
It is shown that FSS is not suitable for an upscaling of the production process and PR is introduced as an alternative and an approach of transferring the material production from the FSS to the PR, without changing the product properties significantly. Pre-existing mathematical PR models were reviewed and their advantages and shortcomings were discussed and a hybrid approach featuring a one-dimensional and a three-dimensional model idea was introduced, but a detailed description of it was not presented. The measurement setup at a laboratory scale PR was presented and related knowledge that could be gained by the measurements for a deeper understanding of the involved processes in order to validate the mathematical models, is discussed.
The article is of interest of Sustainability journal, but specific aspects mentioned in the following require the revision of the paper:
- Only the proof of concept is delivered for zirconia and silica via empiric studies;
- Empiric studies and comparation between operation parameters at FSS and PR for synthesis of ZrO2 and SiO2 was presented in table 2 and table 3, but information related to actual parameters used during experiments are incomplete, so that the results obtained cannot be verified (e.g. temperature, pressure, residence time, energy consumption, pulsation frequency etc).
- Theoretical information related to the heat transfer at particles in steady and pulsating flows, were presented in figures 9 to 11 but in order to be able to evaluate whether the proposed model correlates with the experimental data, numerical values obtained should be presented.
Conclusion: the paper is interesting but not convincing. Not enough experimental results were presented. For a superior quality of the paper the experiments should be presented in a more in-depth way.
Author Response
Stefan Heidinger
Technische Universität Dresden
TU Dresden-Boysen-Research Training Group
Chair of Process Engineering
01062 Dresden
25/02/2021
Dear Madam or Sir,
Thank you for your comments on our manuscript “Advancements in Material Production using Pulsation Reactor Technology”. In the following, we will address each of your comments:
1. English language and style are fine/minor spell check required
- The writing of the manuscript was improved, while a professional native speaker proofread the manuscript suggesting many changes, most of which were incorporated.
2. Only the proof of concept is delivered for zirconia and silica via empiric studies
- The proof of concept with two model materials was indeed conducted in an empirical way. An upscaling of the synthesis of more sophisticated materials and a proof of the model with experimental data will be the content of continuing work following the presented approach.
3. Empiric studies and comparison between operation parameters at FSS and PR for synthesis of ZrO2 and SiO2 was presented in table 2 and table 3, but information related to actual parameters used during experiments are incomplete, so that the results obtained cannot be verified (e.g. temperature, pressure, residence time, energy consumption, pulsation frequency etc).
- All requested parameters have been added to Table 2 and Table 3. Additionally, important FSS synthesis parameters like the combustion and dispersion gas flows, as well as the dispersion pressure drop were added. Further parameters like temperature and residence time in the FSS are difficult to assess in this setup, so an estimation was retrieved from references 11 and 12. The pressure is ambient.
4. Theoretical information related to the heat transfer at particles in steady and pulsating flows, were presented in figures 9 to 11 but in order to be able to evaluate whether the proposed model correlates with the experimental data, numerical values obtained should be presented.
- The heat transfer at particles was considered qualitatively, while applying very basic, well-established and reliable relations. The application of specific and sophisticated models for heat transfer, delivering a quantitative estimate of the Nusselt number, was omitted on purpose. This was conducted in order to focus on the phenomenon of enhanced heat and mass transfer in pulsating flows itself, which is well documented in literature (some of it cited in the manuscript). Future research will provide a sophisticated model with numerical results, which indeed needs validation through experimental data. The wording in the respective section was adjusted in order to avoid confusion.
5. Additional Improvements
- The title of the manuscript was changed to “Material Treatment in the Pulsation Reactor - From Flame Spray Pyrolysis to Industrial Scale” in order to better display the focus of the manuscript not only on pulsation reactor technology, but rather the transfer of the synthesis process from the flame spray synthesis in the lab to the pulsation reactor for mass production.
- The organization of the figures was improved, ensuring that the figures were always referenced before actually appearing. Some figures were corruptly displayed due to formatting issues.
- The format of all sources was revised, with some sources being updated in order to provide all necessary details.
Thank you for helping to improve this manuscript.
Sincerely,
Stefan Heidinger
Round 2
Reviewer 1 Report
This manuscript has been improved after the revision and can be considered to accept.
Reviewer 3 Report
-